# Thermosensitive Chitosan-Containing Hydrogels: Their Formation, Properties, Antibacterial Activity, and Veterinary Usage

**DOI:** 10.3390/gels8020093

**Published:** 2022-02-04

**Authors:** Natalia O. Gegel, Anna B. Shipovskaya, Zaur Yu. Khaptsev, Roman V. Radionov, Anastasia A. Belyaeva, Vitaly N. Kharlamov

**Affiliations:** 1Department of High-Molecular-Weight Compounds, Saratov State University Named after N.G. Chernyshevsky, Astrakhanskaya St., 83, 410012 Saratov, Russia; gegelno@yandex.ru (N.O.G.); miss-n-belyaeva@yandex.ru (A.A.B.); harlamovvitalik4@gmail.com (V.N.K.); 2Microbiology, Biotechnology and Chemistry, Saratov State Vavilov Agrarian University, Sokolovaya St., 335, 410005 Saratov, Russia; dfst@list.ru; 3Animal Science and Veterinary, Michurian State Agrarian University, International St., 110, 393760 Michurinsk, Russia; roman5875@mail.ru

**Keywords:** chitosan hydrochloride, Pluronic F-127, thermosensitive hydrogels, antibacterial activity, veterinary drugs

## Abstract

Mixtures of aqueous solutions of chitosan hydrochloride (CS·HCl, 1–4 wt.%) and Pluronic F-127 (Pl F-127, 25 wt.%) were studied using vibrational and rotational viscometry; the optimal aminopolysaccharide concentration (3 wt.%) and the CS·HCl:Pl F-127 ratio (30:70) to obtain a thermosensitive hydrogel were found. It was shown that at 4 °C, such mixed compositions were viscous liquids, while at 37 °C for 1–2 min, they undergo a thermally reversible transition to a shape-stable hydrogel with a developed level of structure formation, satisfactory viscosity and high mucoadhesive parameters (maximum pull-off force *F*_max_ = 1.5 kN/m^2^; work of adhesion *W* = 66.6 × 10^−3^ J). Adding D-ascorbic acid to the hydrogel led to orientational ordering of the supramolecular structure of the mixed system and significantly improved mucoadhesion (*F*_max_ = 4.1 kN/m^2^, *W* = 145.1 × 10^−3^ J). A microbiological study revealed the high antibacterial activity of the hydrogel against Gram-negative and Gram-positive bacterial strains. The treatment of mixed bacterial infection in cows demonstrated the possibility of the in situ formation of a viscoelastic gel and revealed its high therapeutic effect. It has been suggested that our thermosensitive mucoadhesive CS·HCl:Pl F-127 hydrogels could be considered as independent veterinary drugs and pharmaceuticals.

## 1. Introduction

Nowadays, novel pharmaceutical hydrogel forms to deliver drugs to areas of difficult localization are actively sought for. To solve this problem, “smart” polymer hydrogels that undergo a liquid–hydrogel phase transition depending on temperature, pH or ionic strength of the medium can be considered [1]. The most promising are systems which are in the state of a viscous polymer sol or a viscous polymer liquid at room temperature but undergo gelation to form a viscoelastic composition at the physiological temperature of a living organism. Such thermosensitive hydrogels have unique advantages, including the ability to inject the drug into an inaccessible localization site, local delivery and dose reduction in the drug administered, as well as high adhesion to the mucous surfaces of native tissues [2,3,4].

The aminopolysaccharide chitosan can be considered a promising polymer for the production of biomedical hydrogel preparations, which, when dissolved in an aqueous acidic medium (the formation of a salt form), acquires a positive charge. This polymer is biocompatible with the tissues of a living organism, causes no systemic pathological processes, is biodegradable in a metabolic environment, and has antibacterial and antiviral activity, mucoadhesion [5,6,7,8]. However, chitosan exhibits no ability of gelation, in particular, to form thermo- and pH-sensitive hydrogels, without its chemical modification or the use of cross-linking reagents [4,9,10,11,12,13].

Recently, poloxamers (triblock copolymers of polyethylene oxide and propylene oxide), known under the trademark of Pluronic™ [14,15,16,17], have been used as a gelation component to obtain “smart”, thermosensitive, chitosan-containing hydrogels. Aqueous solutions of Pluronics are characterized by a lower critical solution temperature, below which no phase separation is observed in the system at any polymer concentration, and they pass into a gel-like state with an increase in temperature [18]. Nevertheless, despite these attractive properties of Pluronics, thermosensitive hydrogels based thereon have weak mechanical strength due to a decrease in viscosity under shear deformation, exhibit almost no mucoadhesion and quickly dissolve at the injection site, which limits their biomedical application [19,20].

Combination with mucoadhesive polymers, in particular, chitosan, can be a way to improve the rheological and adhesive properties of Pluronics [4,21]. It is known from the literature [16,22] that Pluronic has good compatibility with this aminopolysaccharide. The introduction of chitosan into an aqueous solution of Pluronic promotes the formation of thermosensitive mucoadhesive hydrogels in situ in a wide temperature range, provides the optimal viscosity for uniform distribution of the hydrogel at the application site and high mucoadhesion to the mucous tissue surface, and improves the mechanical and textural properties [4,23]. Small additions of chitosan to the poloxamer change the coefficients of diffusion in the gel structure, which contributes to the accommodation of unbound water resulting from dehydration of the Pluronic micelle core, gelation improvement and an increased viscosity [21]. Mixtures of chitosan with the amphiphilic block copolymer Pluronic F-127 (Pl F-127) are the most studied from both medical and practical points of view. This copolymer belongs to the class of nonionic surfactants [4,16,24,25]; however, only small additions of the aminopolysaccharide were used in the studies, not exceeding 1.5–1.6 wt.%.

Like chitosan, synthetic amphiphilic Pluronic block copolymers have a number of biologically useful properties [16]. They are used as immunoadjuvants [26], in the treatment of burns [27] and in postoperative wound care [28]. In addition, Pluronics are able to reduce multidrug resistance, as well as to protect living cells from mechanical damage, increasing the modulus of elasticity of the cell membrane, and, in the event of such damage, restoring its integrity [25,29,30]. In combination with biologically active chitosan, Pluronic-based hydrogels are used as injection-implantable gel systems and matrices for transdermal, transconjunctival and transmucosal drug delivery [21,23,24,31,32,33].

According to our previous studies, water-soluble salt chitosan in the form of hydrochloride with biologically active additives, for example, D-ascorbic acid (D-AscA), is the most promising for medico-pharmaceutical use. The use of an aqueous D-AscA solution for dissolving CS·HCl allows one, on the one hand, to increase the number of protonated amino groups [34], and on the other hand, to combine the biochemical activities of the aminopolysaccharide macromolecule and the acid anion with a synergistic effect. Hydrogel materials based on such a system exhibit high antibacterial activity and also accelerate the proliferation of epithelial and epithelial-like cells in vitro [35,36]. In an in vivo study of wound healing activity, a higher rate of healing of burn wounds was observed due to alternative activation of tissue macrophages and the suppression of the production of inflammatory mediators [37,38]. In addition, they have immunocorrective properties to activate local immunity [37]. It seems that the use of CS·HCl in combination with a Pluronic, in particular Pl F-127, and an additive of D-AscA could be very promising for the development of new, highly effective hydrogel pharmaceuticals without the introduction of drugs into them. The above predetermined the goal of this study.

The aim of this work was to study the rheological and adhesive properties of mixtures of aqueous solutions of CS·HCl and Pl F-127 without and with the addition of D-AscA to obtain thermosensitive hydrogels, undergoing gelation at the physiological temperature, with a high content of aminopolysaccharide and high antibacterial activity for their prospective use in veterinary medicine.

## 2. Results and Discussion

### 2.1. Rheological Tests

In the first stage, the temperature dependence of the moduli of elasticity and loss of the aqueous CS·HCl:Pl F-127 mixture was studied to find the optimal concentration of the initial CS·HCl solution to obtain a thermosensitive hydrogel. Figure 1 shows the dependences G’(*T*) and G”(*T*) of the CS·HCl:Pl F-127 system in a 30:70 volume ratio at *C*_Pl F-127_ = const and *C*_CS·HCl_ = 1–4 wt.%. According to Wanka G. et al. [18], the content of 70 vol.% of a Pl F-127 aqueous solution with the concentration used in the work in the mixed composition is sufficient for the liquid–hydrogel phase transition in the temperature range of 30–40 °C.

As is seen from Figure 1, the moduli of elasticity and loss of the studied CS·HCl:Pl F-127 mixtures at 4 °C have similar values, are relatively low and do not depend on *C*_CS·HCl_ in the mixed composition. This behavior is typical for viscous liquids, which almost dnever exhibit elastic properties, and indicates an insignificant contribution of the elastic component to the phase state of the system. As the temperature rises, the values of G’ and G” increase. The concentration of CS·HCl in the mixture has a significant effect on the character of the dependences G’(*T*) and G”(*T*). E.g., at 25 °C, the largest value of the elasticity modulus takes place at *C*_CS·HCl_ = 2 wt.%. Attention is drawn to a significant increase in this modulus compared to the loss modulus, indicating the predominance of the elastic component and the beginning of the transition of the system to a gel-like state. At 37 °C, the same character of G’ and G” with the predominance of the elastic component (G’ > G”) is observed for the composition with *C*_CS·HCl_ = 3 wt.%. At *C*_CS·HCl_ = 4 wt.%, the dependences G’(*T*) and G”(*T*) almost coincide with those for *C*_CS·HCl_ = 1 wt.%, which may be due to the destruction of the Pluronic micellar structure in an acidic medium due to the hydration of the nuclei of micellar clusters or the increase in the proportion of non-aggregated copolymer (unimeric) chains [39,40]. Therefore, at 25 and 37 °C, the mixed compositions with *C*_CS·HCl_ = 2 and 3 wt.% are viscoelastic systems with the predominance of elastic properties over viscous ones. Based on the results obtained, the CS·HCl:Pl F-127 mixtures with *C*_CS·HCl_ = 3 wt.% (i.e., with the highest content of the biologically active aminopolysaccharide and undergoing gelation at the phisiological temperature) were selected for further studies.

In the next stage, to find the optimal content of Pl F-127 in the composition, the viscosity properties of CS·HCl:Pl F-127 mixtures were studied by varying the ratio of the components. Our evaluation of the rheological behavior showed that the viscosity rheograms of the initial 3 wt.% CS·HCl solutions within the entire temperature range were described by the logη = *f* (logτ) dependences which are classical for polymeric systems, with regions of the highest Newtonian and structural viscosity (Figure 2a–c; curves *1*), those of 25 wt.% Pl F-127 solutions showed flow curves of dilatant liquids at 4 °C (Figure 2a; curve *5*) and high-viscosity structured systems with ultimate strength at 25 and 37 °C (Figure 2b,c; curves *5*). The mixed CS·HCl:Pl F-127 compositions in volume ratios of 70:30 and 50:50 within 4–37 °C, like CS·HCl, showed a flow character typical for viscous-flow polymeric systems (Figure 2a–c; curves *2* and *3*). The viscosity properties of the CS·HCl:Pl F-127 = 30:70 (vol./vol.) system were similar to a solution of the individual Pluronic and differed in lower viscosity values alone (Figure 2a–c; curves *4*). The rheological properties of the mixture of this composition at 25 and 37 °C, as well as the dependences G’(*T*) and G”(*T*) in Figure 1 (curves *3*), testify to the predominance of elastic deformation over viscous flow in the system and the formation of a spatial hydrogel network in the range of 25–37 °C. When comparing the viscosity rheograms with the moduli of elasticity and loss, it is not difficult to see that at 37 °C, the hydrogel system was characterized by mechanical strength and high elasticity (logη ~ 3.5 [mPa∙s] at logτ = 2.7 [mPa∙s], G’ > G”, G’/G” ~ 2.8). At 25 °C, despite the high values of viscosity, a “loose”, gel-like structure was formed; although it had some strength and the ability of elastic deformation, it almost did not show elasticity and was easily destroyed under shear deformation conditions (logη ~ 3.4 [mPa∙s] at logτ = 2.7 [mPa∙s], G” > G’, G”/G’ ~ 1.9).

Based on the data obtained, the dependence of the maximum viscosity (η_max_) of CS·HCl:Pl F-127 mixtures on the volume fraction of Pl F-127 in the composition was plotted using the Vinogradov–Pokrovsky model (Figure 3a). With an increase in the content of Pl F-127, the viscosity of the mixtures increased at all temperatures. At 4 °C, the increase in η_max_ was insignificant and, starting from the composition CS·HCl:Pl F-127 = 50:50 (vol./vol.), the viscosity values did not depend on the ratio of the components (curve *1*). At 25 and 37 °C, starting from the ratio CS·HCl:Pl F-127 = 50:50 (vol./vol.), η_max_ increased by almost three and four orders of magnitude, respectively (curves *2* and *3*). Based on the oscillatory and rheological characteristics, the CS·HCl:Pl F-127 30:70 (vol./vol.) system seemed to be optimal for obtaining a hydrogel, as it exhibited a high level of structure formation and contained the minimum amount of Pl F-127 required for gelation. This mixture was liquid at 4 °C, and upon rapid heating up to 37 °C, it underwent a transition into a shape-stable hydrogel (Figure 3b). In addition, the structured liquid–hydrogel transition was thermally reversible and occurred in as little as 1–2 min.

### 2.2. Mucoadhesive Properties

Figure 4 shows the results of our mechanical tests evaluating the adhesive properties of hydrogels to the mucin surface, the adhesion parameters are in Table 1. The choice of mucin as a model substrate was due to the fact that it is the main mucopolysaccharide of the mucous membranes of living tissues [41]. The dependence *F* = *f*(*S*), as well as the values of *F*_max_ and *W* for the CS·HCl:Pl F-127 = 30:70 (vol./vol.) hydrogel indicate its good mucoadhesive ability (curve *1*). The obtained adhesive characteristics of the hydrogel were significantly higher than those of individual CS [6,8], and are also described in the literature for its mixtures with syntetic polymers [7], in particular, poloxamers [4,21,23].

Since individual Pluronic hydrogels are characterized by relatively weak adhesion and are incapable of forming strong bonds with mucin [19,20], the high mucoadhesive properties of our CS·HCl:Pl F-127 mixtures were due to the presence of chitosan.

### 2.3. Effect of D-AscA on Rheological and Mucoadhesive Properties

The addition of D-AscA to the mixed composition with CS·HCl:Pl F-127 = 30:70 (vol./vol.) had almost no effect on the values of the moduli of elasticity and loss at 4 °C, as well as on G’ at 25 °C; it slightly lowered G’ at 37 °C and significantly lowered G” at 25 and 37 °C (Figure 1, curves *5*). As for the source mixture, an increase in temperature above 25 °C was accompanied by a sharp increase in G’; however, the ratio of the components of the complex modulus for the composition with the biologically active additive was much higher (G’/G” ~ 9.4). This character of the G’(*T*) and G”(*T*) dependences indicates the formation of a more ordered and more elastic hydrogel structure.

Our study of the rheological properties of the mixed CS·HCl:Pl F-127 + D-AscA composition at 4 °C showed a decrease in viscosity and a change in the nature of the viscous flow from a dilatant liquid to a weakly structured polymeric one (Figure 2a, curve *6*). In the range of 25–37 °C, the viscosity values were significantly higher compared to the source composition, and the logη = *f*(logτ) dependences were typical for gel-like systems with a high degree of structuring (Figure 2b,c, curve *6*). The values of η_max_ are comparable with those for individual Pl F-127.

The addition of D-AscA significantly increased the strength of the adhesive bond of the hydrogel system to the model substrate, which was confirmed by the increased *F*_max_ and *W* values (Figure 4, curve *2*; Table 1). The adhesive characteristics of the chitosan-containing hydrogel with D-AscA became comparable to those of the mucoadhesive hydrogel Metrogyl Denta™ used in the treatment of periodontal and oral mucosa diseases (Figure 4, curve *3*).

Adhesion chitosan–mucin contacts are known to be formed through electrostatic interactions between the positively charged amino groups of chitosan and the negatively charged sialic acid residues of mucins, as well as hydrophobic interactions of the methyl groups of acetylated chitosan residues with the methyl groups of mucin side chains [6,21,39]. The CS·HCl sample used in this work contained ~18 mol% of free –NH_2_ groups available for protonation even with weak acids, which included D-AscA, and ~20 mol% of acetylated amino groups, providing a high level of hydrophobic contacts of the –NHCOCH_3_ groups of chitosan with the –CH_3_ groups of mucin and high adhesion. According to our previous studies, the addition of D-AscA to an aqueous solution of the CS·HCl sample used in this work increased the number of protonated amino groups [34], which, in turn, increased the number of electrostatic contacts and, accordingly, enhanced the strength of the mucoadhesive bond of the hydrogel with the model substrate.

### 2.4. Scanning Electron Microscopy (SEM)

Our examination of the solid phase of the CS·HCl:Pl F-127 = 30:70 (vol./vol.) hydrogel showed that the sample had an almost flat surface relief and a homogeneous internal structure with no signs of phase separation (Figure 5a). This, like the results of refs [16,22], indicates the compatibility of the components of the mixed composition. The addition of D-AscA to the thermosensitive hydrogel did not affect the compatibility of the components; however, it led to ordering of the supramolecular organization of the polymer mixture (Figure 5b). The SEM data confirm the results of oscillatory and rotational viscometry.

### 2.5. Antibacterial Activity In Vitro

Our microbiological study of the antibacterial activity of thermosensitive CS·HCl:Pl F-127 = 30:70 (vol./vol.) hydrogels without and with the addition of D-AscA showed that both compositions exhibited high antibacterial activity against Gram-negative (*P. aeruginosa* and *S. typhimurium*) and Gram-positive (*B. cereus* and *S. aureus*) bacterial strains (Figure 6, Table 2). The largest clearing zones were observed for *P. aeruginosa* and *S. aureus*, which may be associated with deeper destructive changes in their cellular structures in comparison with *S. typhimurium* and *B. cereus*. The diameter of the microorganism growth inhibition zones on our chitosan-containing hydrogels was comparable to that for aqueous solutions of CS·HCl + D-AscA and was significantly larger than for individual D-AscA. Hence, it follows that the D-AscA additive enhances the antimicrobial effect of CS·HCl, which is retained in the mixed CS·HCl:Pl F-127 composition.

Despite the high bacteriostatic effect, the addition of D-AscA slightly reduced the antibacterial activity of our thermosensitive CS·HCl:Pl F-127 = 30:70 (vol./vol.) hydrogel. Meanwhile, we previously established a significant increase in the effectiveness of the treatment of patients with chronic generalized periodontitis against the background of the use of our CS·HCl + D-AscA hydrogel, which has a significantly lower antibacterial effect than most antiseptics [37]. It was shown that standard antiseptic therapy eliminates subgingival and supragingival pathogenic microflora only, while the high therapeutic effect of CS·HCl + D-AscA is achieved due to the immunotropic effect on the innate immunity effectors as well. A similar immunocorrective activity should be expected from our CS·HCl:Pl F-127 + D-AscA hydrogel.

### 2.6. Therapeutic Effect In Vivo

The therapeutic effect was estimated in the treatment of vaginitis (mixed bacterial infection) in cows using the thermosensitive CS·HCl:Pl F-127 = 30:70 (vol./vol.) hydrogel with the addition of D-AscA, enclosed in a gelatin matrix in the form of a suppository. It was shown in special experiments that heating the suppository up to 36–40 °C leads to the transition of gelatin into a viscous-flowing state, while the CS·HCl:Pl F-127 mixture passes into a shape-stable hydrogel. Under the conditions of a living organism (37–39 °C), the formation of a viscoelastic gel in situ was observed.

In the experimental group, when using our suppository, the positive dynamics of animal recovery was observed already on the first day of treatment and reached ~20% (Figure 7a). On the second day, the number of recovered animals in the experimental group was ~60%, whilst in the control group, this value was only ~20%. Full recovery (~100% of recovered animals) was observed on the third day of treatment, whilst in the control group, this was only seen on the fifth day.

Figure 7b–e shows photos with the clinical picture of traumatic postpartum vaginitis before and after treatment with our suppositories and Ichthyol IUDs. In the acute phase of the disease, an increase in the genitals of the animal is observed due to edema and hyperemia of the mucous membrane, as well as localized areas of purulent exudate (Figure 7b,d). After 3 days of the treatment with our suppositories, the tissues acquire a pale pink color; the mucous discharge becomes transparent (Figure 7c). With traditional therapy, a similar therapeutic picture was only observed on the fifth day of treatment (Figure 7e).

In general, the conducted assessments of the in vivo pharmacological activity of the test drug in the treatment of serous-catarrhal vaginitis in cows have revealed a significantly higher therapeutic effect compared to standard treatment [42]. Due to the biological activity of chitosan and Pl F-127 [5,16], the use of our thermosensitive hydrogel based thereon imposes no restrictions on the use of animal milk and meat for food purposes. In addition, this drug can be used for the prevention of bacterial infections in veterinary practice.

## 3. Conclusions

Our rheological analysis of the viscoelastic properties of mixtures of aqueous solutions of chitosan hydrochloride and Pluronic with the varied concentration of CS·HCl and CS·HCl:Pl F-127 ratio revealed the optimal compositions for obtaining a mucoadhesive hydrogel with a high content of the aminopolysaccharide and undergoing a liquid–hydrogel transition at a living organism’s temperature in situ. The addition of D-AscA to the chitosan-containing hydrogel did not affect the compatibility of the components, led to ordering of the supramolecular structure of the mixture and significantly increased its mucoadhesion parameters, which became comparable to those of commercial mucoadhesive preparations. The resulting hydrogel exhibited high antibacterial activity against Gram-positive and Gram-negative strains of microorganisms. The pharmacological activity of our thermosensitive hydrogel in the treatment of mixed bacterial infection of animals was significantly higher than that of traditional antibacterial drugs. This is important, since high antimicrobial activity and a therapeutic effect were achieved due to the intrinsic biological activity of the components. It can be stated that CS·HCl:Pl F-127 mixtures are very promising in the development of new, highly effective pharmaceutical hydrogel preparations not only for veterinary purposes, but also for the treatment of diseases of human mucous membranes.

## 4. Materials and Methods

### 4.1. Materials

The following reagents were used: CS·HCl with the molar fractions: (NH_2_·HCl) − 0.62, (NH_2_) − 0.18, (NHCOCH_3_) − 0.2 with a viscosity average molecular weight 38 kDa, a degree of deacetylation DD = 80 mol % (Bioprogress Ltd., Shchelkovo, Russia); the copolymer of poly(propylene oxide) and poly (ethylene oxide) Pl F-127 with a molecular weight 12.6 kDa (Sigma-Aldrich, St. Louis, MO, USA); D-AscA with 98% basic substance (Khimreaktiv Corp., Moscow, Russia); distilled water; Mucin with 99% basic substance—mucous secretion filtrate, Helix Aspersa Snail (La Coruña, Spain); gelatin (Khimreaktiv Corp., Moscow, Russia). All reagents were chemical grade and used without further purification.

### 4.2. Solution Preparation

Aqueous solutions of CS·HCl with concentrations *C*_CS·HCl_ = 1–4 wt.% (as well as with the addition of 3 wt.% D-AscA) were prepared by dissolving a weighed portion of the polymer powder (CS·HCl and D-AscA powders) in distilled water at ~22 ± 2 °C under normal atmospheric pressure for 24 h. Aqueous solutions of Pl F-127 with the concentration *C*_Pl F-127_ = 25 wt.% were prepared by dissolving a weighed portion of the copolymer in distilled water at 4 °C, at which the Pl F-127 + H_2_O system is in a liquid state [18], and stored in a refrigerator at 4 °C. CS·HCl:Pl F-127 compositions were prepared by mixing the source CS·HCl solutions without and with the addition of D-AscA (~22 ± 2 °C) and Pl F-127 (4 °C) at room temperature in volume ratios (vol./vol.) 70:30–30:70, and stored at 4 °C.

### 4.3. Rheological Tests

Viscosity rheograms logη = *f* (logτ), where η is the viscosity (mPa∙s) and τ is the shear stress (Pa), were recorded on a Rheotest RN 4.1 rotational viscometer (Germany) with a cylinder–cylinder working unit (inner cylinder H1) in the range logτ = 0.1–3.0 [Pa]. The highest viscosity values (η_max_) of concentrated systems, for which no Newtonian flow region was experimentally fixed, were calculated using the modified Vinogradov–Pokrovsky rheological model using the wxMaxima program [43]. The viscoelastic properties of the system were evaluated in an oscillatory mode with a cone–cylinder measuring system (inner cone S2). The moduli of elasticity (G’) and loss (G”) were measured at a constant amplitude of 0.01 mN·m in the frequency range of 0.5–10 Hz. Rheological tests were carried out at 4, 25 and 37 °C, the thermostating time was 30 min.

### 4.4. Adhesive Properties

Adhesion properties were assessed in vitro on an Instron 5965 tensile testing machine (England) under conditions of overcoming the adhesion forces of the test sample from its interface with a model substrate (mucin) at 37 °C according to our modified method for thermosensitive hydrogels [21,44]. Two filter paper discs, 5 cm in diameter, one of which was covered with a mucin layer (~22 ± 2 °C), the other covered with a layer of the test sample (4 °C), were heated up to 37 °C, fixed on the stationary and movable cylindrical platform, respectively, and using double-sided adhesive tape, they were brought into contact with a force of 2.5 N for 3 min to form an adhesive bond, and a peeling force was applied at a constant speed of 1 mm/min until the contacting surfaces were completely detached. The force was recorded as a function of the displacement *F* = *f*(*S*), from which the maximum pull-off force *F*_max_ was determined as the maximum force of the positive peak reduced to the area of the contacting surfaces (2 μm^2^), and the work of adhesion *W* was calculated as the area under the *F* = *f*(*S*) curve [45]. A commercial gum gel Metrogyl Denta™ (Unique Pharmaceutical Laboratories, India) was used as a comparison sample. Five parallel experiments were carried out. Statistical analyses were carried out using the SPSS software package (SPSS Statistical Software, Inc., Chicago, IL, USA).

### 4.5. Scanning Electron Microscopy

The surface morphology of the solid phase of our hydrogels was explored via SEM on a MIRA \\ LMU microscope at a voltage of 8 kV and a conducting current of 60 pA. The sample was prepared as follows: mixed CS·HCl:Pl F-127 compositions without and with the addition of D-AscA were kept at 37 °C until a shape-stable hydrogel was formed, placed into ethyl alcohol for 30 min, then dried in air for 24 h. Before carrying out microscopic studies, a 5 nm thick layer of gold was deposited onto the obtained sample using a K450X Carbon Coater.

### 4.6. Antibacterial Activity

Antibacterial activity was assessed in vitro via the agar diffusion method using the model of Gram-negative *Pseudomonas aeruginosa* (ATCC 9027) and *Salmonella typhimurium* (non-collectible) and Gram-positive *Bacillus cereus* (NCTC 8035) and *Staphylococcus aureus* (ATCC 6538) strains cultured in a meat peptone. A suspension of cells of a daily bacterial test culture (seeding dose of 50 million microbial bodies / mL) was introduced into melted meat-peptone agar (50 ± 1 °C), poured into glass Petri dishes and left to gelate. Four wells with a diameter of 10 mm each were made in the frozen meat-peptone agar around each Petri dish’s circumference at a distance of 20–25 mm from the center, 100 μL of the test CS·HCl:Pl F-127 sample was added into two wells (parallel experiments), and CS·HCl:Pl F-127 with the addition of D-AscA (4 °C) were added in the other two wells (parallel experiments as well) and quickly heated up to 37 °C. The bacterial growth inhibition zones were measured at 37 °C after 16–18 h of cultivation. The experiment was repeated three times. Similar studies were carried out for aqueous solutions of CS·HCl + D-AscA and individual D-AscA (control experiments).

### 4.7. Treatment and Prevention of Vaginitis in Cows

Therapeutic efficacy was assessed in vivo during the treatment of serous-catarrhal vaginitis in mature black-finger cows at the ages of 3–4 years after calving (collective farm Zarya, Tamalinsky district, Penza region, Russian Federation). The experimental group of animals was treated with the use of suppositories from our test drug at a dosage of one suppository once a day; the control group was treated with the traditional veterinary drug “Intrauterine sticks with ichthyol” (CJSC NPP Agrofarm, Russian Federation) according to the instructions. Each group consisted of 5 cows. Observation was carried out until the animals recovered. The dynamics of recovery was assessed according to the following clinical signs: a decrease in exudative discharge, transparency and consistency of the exudate.

The suppository was prepared as follows. A solution of CS·HCl:Pl F-127 = 30:70 (vol./vol.) with *C*_CS·HCl_ = 3 wt.% and the addition of 3 wt.% D-AscA (4 °C) was mixed with a 13% aqueous gelatin solution (60 °C), stirred with a glass rod, poured into a silicone cylindrical mold with a pointed end 5 cm long and a maximum diameter of 1.5 cm, cooled down to room temperature and stored at 4 °C. The in vitro and in situ transition of the suppository from the solid state to the hydrogel form was observed on a heating table with a temperature control and regulation sensor (RF) and using a veterinary vaginal speculum, respectively. The photos were taken with a digital camera Canon PowerShot G7 X Mark II.

## 5. Patents

Kharlamov, V.N.; Gegel, N.O.; Shipovskaya, A.B.; Khaptsev, Z.Y.; Rodionov, R.V. Means for the prevention and treatment of vaginitis in cows. Patent RF No. 2751876. 2021.

## Figures and Tables

**Figure 1 gels-08-00093-f001:**
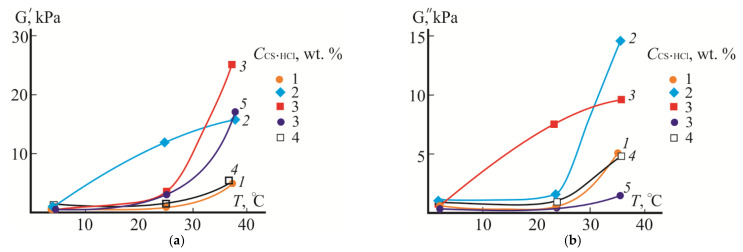
Temperature dependences of the moduli of elasticity (**a**) and loss (**b**) of the aqueous CS·HCl:Pl F-127 = 30:70 (vol./vol.) mixture with *C*_CS·HCl_ = 1–4 wt.%, *C*_Pl F-127_ = 25 wt.% without (1–4) and with the addition of 3 wt.% D-AscA (*5*).

**Figure 2 gels-08-00093-f002:**
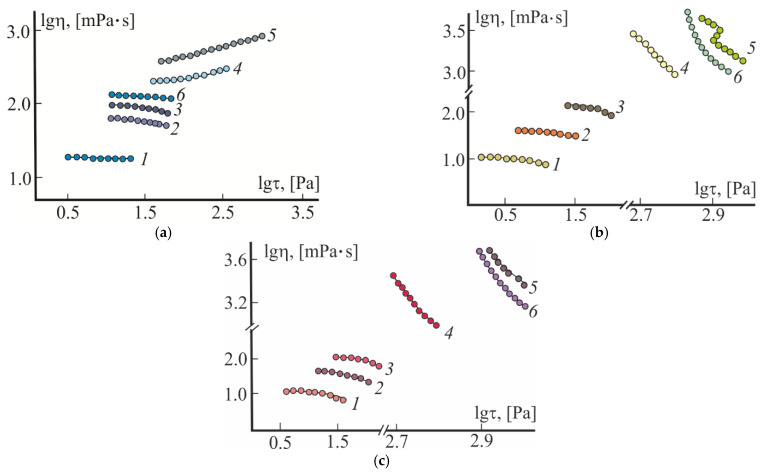
Viscosity curves of aqueous CS·HCl solutions with *C*_CS·HCl_ = 3 wt.% (*1*), Pl F-127 with *C*_Pl F-127_ = 25 wt.% (*5*) and CS·HCl:Pl F-127 mixtures in volume ratios (vol./vol.): 70:30 (*2*), 50:50 (*3*) and 30:70 (*4*, *6*) without (*2*–*4*) and with an additive of 3 wt.% D-AscA (*6*) at 4 (**a**), 25 (**b**) and 37 °C (**c**).

**Figure 3 gels-08-00093-f003:**
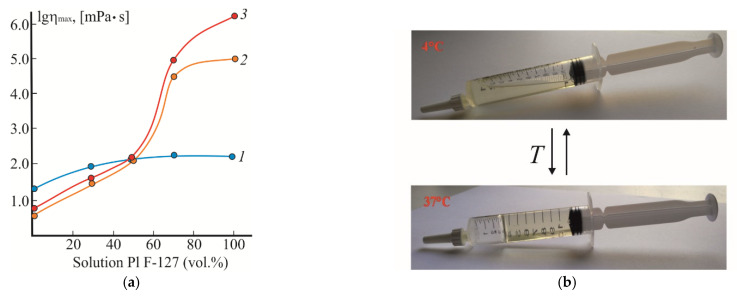
(**a**) Dependence of the highest Newtonian viscosity of the same CS·HCl and Pl F-127 solutions and CS·HCl:Pl F-127 mixtures, as in Figure 2, on the volume fraction of Pl F-127 at 4 (*1*), 25 (*2*) and 37 °C (*3*). (**b**) Photos of the phase state of the CS·HCl:Pl F-127 = 30:70 (vol./vol.) mixture with *C*_CS·HCl_ = 3 wt.% and *C*_Pl F-127_ = 25 wt.% when temperature changes.

**Figure 4 gels-08-00093-f004:**
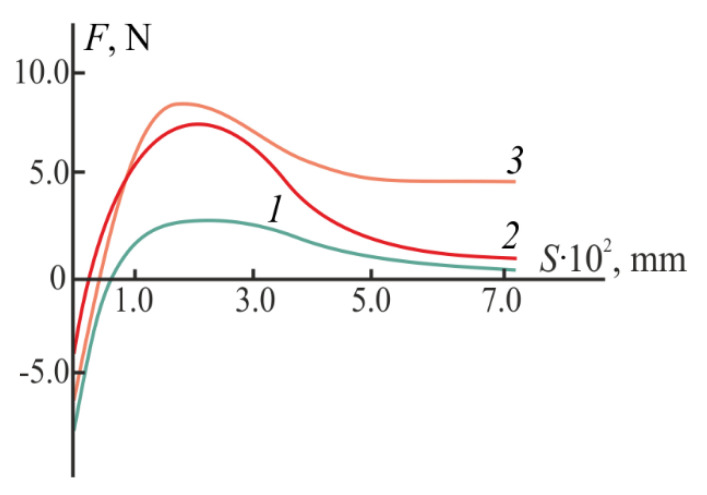
Force–distance dependence for the CS·HCl:Pl F-127 = 30:70 (vol./vol.) hydrogels with *C*_CS·HCl_ = 3 wt.%, *C*_Pl F-127_ = 25 wt.% without (*1*) and with the addition of 3 wt.% D-AscA (*2*) and Metrogyl Denta™ (*3*).

**Figure 5 gels-08-00093-f005:**
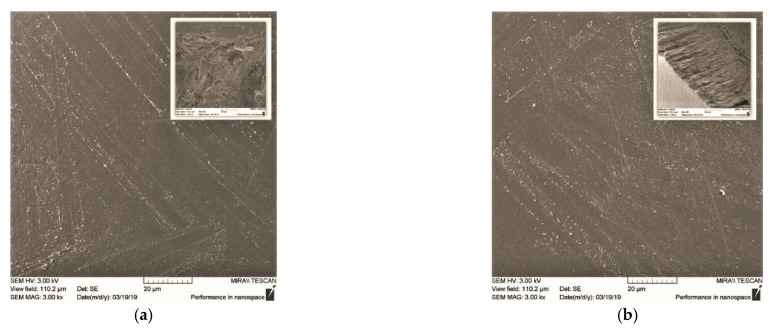
SEM photos of the surface and internal structure (inserts) of the solid phase of the CS·HCl:Pl F-127 = 30:70 (vol./vol.) hydrogels with *C*_CS·HCl_ = 3 wt.%, *C*_Pl F-127_ = 25 wt.% without (**a**) and with the addition of 3 wt.% D-AscA (**b**).

**Figure 6 gels-08-00093-f006:**
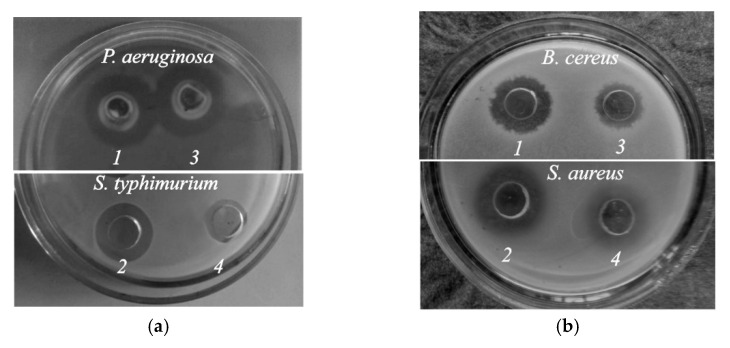
Growth inhibition zones of the test culture *P. aeruginosa* ATCC 9027, *S. Typhimurium* and *B. cereus* NCTC 8035 (**a**), *B. cereus* and *S. aureus* (**b**) by the CS·HCl:Pl F-127 = 30:70 (vol./vol.) hydrogels without (*1*, *2*) and with the addition of 3 wt.% D-AscA (*3*, *4*); *C*_CS·HCl_ = 3 wt.%, *C*_Pl F-127_ = 25 wt.%.

**Figure 7 gels-08-00093-f007:**
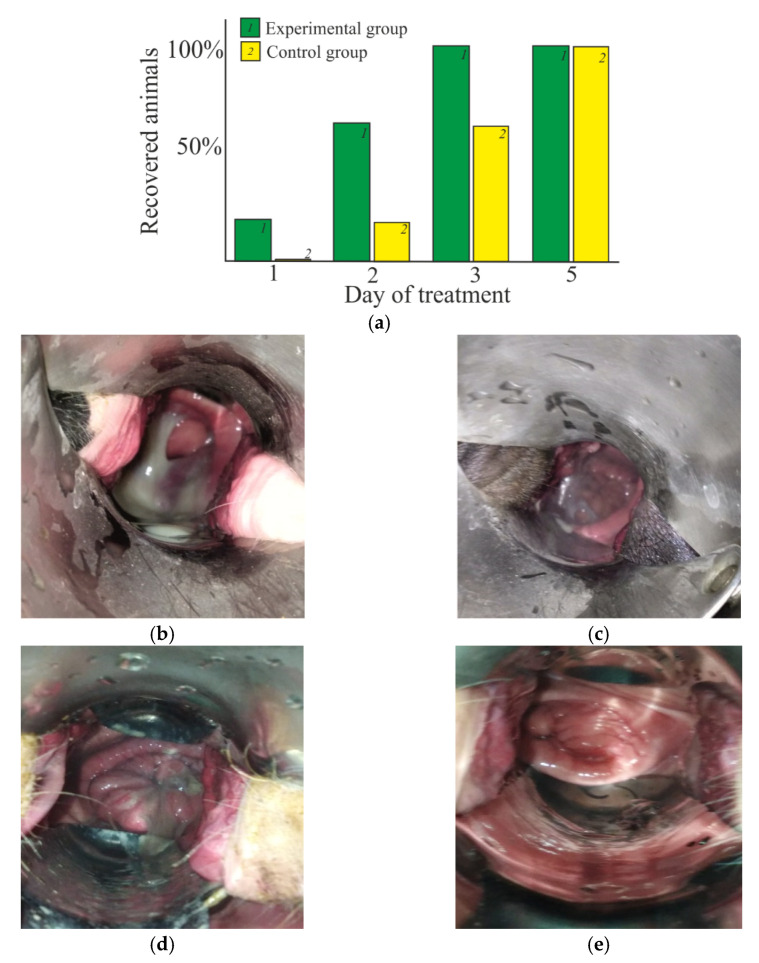
(**a**) Dynamics of the treatment of cows with postpartum serous-catarrhal vaginitis using our CS·HCl:Pl F-127 = 30:70 (vol./vol.) suppositories with *C*_CS·HCl_ = 3 wt.%, *C*_Pl F-127_ = 25 wt.% and the addition of 3 wt.% D-AscA and Ichthyol IUDs (conventional therapy). (**b**–**e**) Clinical picture of traumatic postpartum vaginitis before (**b**,**d**) and after 3 days of the treatment with our suppositories (**c**), after 5 days of the treatment with Ichthyol IUDs (**e**).

**Table 1 gels-08-00093-t001:** Adhesive characteristics of the CS·HCl:Pl F-127 = 30:70 hydrogels with *C*_CS·HCl_ = 3 wt.%, *C*_Pl F-127_ = 25 wt.% without and with the addition of 3 wt.% D-AscA.

Sample	Maximum Pull-Off Force, *F*_max_, N/m^2^	Work of Adhesion, *W*∙10^3^, J
CS·HCl:Pl F-127	1530	66.6
CS·HCl:Pl F-127 + D-AscA	4135	145.1
Metrogyl Denta™	4440	174.3

**Table 2 gels-08-00093-t002:** Antibacterial activity of our thermosensitive CS·HCl:Pl F-127 = 30:70 (vol./vol.) hydrogels with *C*_CS·HCl_ = 3 wt.%, *C*_Pl F-127_ = 25 wt.%. without and with the addition of D-AscA, aqueous solutions of CS·HCl + D-AscA and D-AscA.

Bacterial Test Culture	Bacteriostatic Action				Bacterial Growth Inhibition Zone Diameter, mm			
	CS·HCl:Pl F-127		CS·HCl + D-AscA	D-AscA	CS·HCl:Pl F-127		CS·HCl + D-AscA	D-AscA
	−	D-AscA			−	D-AscA		
*P. aeruginosa* (−)	++	++	++	+	25 ± 1	23 ± 2	27 ± 2	13 ± 1
*S. typhimurium* (−)	++	+	N/A	N/A	20 ± 2	15 ± 2	N/A	N/A
*B. cereus* (+)	++	+	N/A	N/A	21 ± 1	16 ± 1	N/A	N/A
*S. aureus* (+)	++	++	++	+	26 ± 2	24 ± 1	24 ± 2	14 ± 2

++—high antibacterial activity. +—moderate antibacterial activity. N/A—no experiment carried out.

## Data Availability

Data is contained within the article.

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
