# Peer review of "Thermosensitive Chitosan-Containing Hydrogels: Their Formation, Properties, Antibacterial Activity, and Veterinary Usage"

_gels, 2022, doi:10.3390/gels8020093_

Round 1
Reviewer 1 Report
This manuscript describes the preparation of thermosensitive hydrogels by mixing mucoadhesive polymer chitosan with the thermosensitive polymer Pluronic F 127. In addition, D-ascorbic acid was added to the hydrogel and used as a comparison to the primary hydrogels. The hydrogels were examined for their rheology properties, in vitro antibacterial activity, and in vivo anti vaginal infection study. The manuscript was not written well, with too many grammatical errors, which made it inconvenient to read. Here are some of the major comments that need to be addressed.

Author Response
Пожалуйста, смотрите вложение.

Reviewer 2 Report
Dear authors,
I appreciate your effort in designing thermosensitive chitosan-Pluronic F127 hydrogels. Though the research idea is good, you must improve your discussion part for better presentation of results. Please answer the comments to improve the quality of the manuscript.
Thanks and good luck.

Author Response
Пожалуйста, смотрите вложение.

Round 2
Reviewer 1 Report
no comment
Reviewer 2 Report
Article can be accepted